# Effect of Bias Voltage on the Crystal Growth of AlN(002) Thin Films Fabricated by Reactive Magnetron Sputtering

**DOI:** 10.3390/mi16091027

**Published:** 2025-09-08

**Authors:** Yong Du, Haowen Zou, Tiejun Li, Guifang Shao

**Affiliations:** 1School of Ocean Information Engineering, Jimei University, Xiamen 361021, China; duyong2001@jmu.edu.cn (Y.D.); hwzouq@163.com (H.Z.); 2Pen-Tung Sah Institute of Micro-Nano Science and Technology, Xiamen University, Xiamen 361005, China; gfshao@xmu.edu.cn

**Keywords:** AlN, reactive magnetron sputtering, bias voltage

## Abstract

The study investigates the influence of bias voltage on the structural and morphological properties of aluminum nitride AlN (002) thin films deposited on sapphire substrates via reactive magnetron sputtering for high-frequency surface acoustic wave (SAW) devices. The results indicate that applying a positive bias voltage (>0 V) yields AlN films with compact and uniform surfaces. As bias increases, the deposition rate initially rises before declining, while root–mean–square (RMS) roughness progressively decreases, reaching a minimum at 100 V, significantly enhancing surface quality. X-ray diffraction (XRD) analysis reveals enhanced (002) preferential orientation with increasing bias, indicating improved crystallinity. These findings demonstrate that optimized bias voltage not only refines surface morphology but also strengthens crystal alignment, particularly along the (002) plane, making AlN films highly suitable for high-frequency SAW applications, and provides data for the preparation of higher-quality AlN films.

## 1. Introduction

In recent years, research on surface acoustic wave (SAW) devices has been rapidly advancing toward high-end applications such as higher-frequency resonators and high-sensitivity sensors. The operational frequency of SAW devices is intrinsically linked to the propagation velocity of acoustic surface waves on piezoelectric substrates, rendering the preparation of piezoelectric thin films critically important. Piezoelectric films govern the electroacoustic coupling characteristics of SAW sensors, thereby directly influencing their operating frequency and stability. Among the various piezoelectric materials used for SAW devices, AlN thin films with a (002) crystal orientation exhibit excellent surface acoustic wave propagation velocity. This indicates that AlN films with a dominant (002) crystallographic orientation can enable SAW devices to achieve higher operating frequencies, effectively mitigating electromagnetic interference in practical applications [1]. Structurally, (100)-oriented AlN consists of B1 bonds, whereas (002)-oriented AlN comprises both B1 and B2 bonds. A schematic diagram of bond angles in the AlN piezoelectric material is shown in Figure 1. Notably, B2 bonds along the c-axis exhibit longer bond lengths, lower bond energies, and higher susceptibility to fracture. Consequently, precise control of experimental parameters to regulate the energy of sputtered particles is essential for selectively promoting B1 or B2 bond formation, thereby achieving AlN thin films with desired preferential orientations.

The current fabrication techniques for aluminum nitride (AlN) thin films primarily include reactive magnetron sputtering [2,3,4,5,6,7,8,9,10,11,12], plasma-enhanced atomic layer deposition (PEALD) [13,14], and metal–organic chemical vapor deposition (MOCVD) [15,16,17]. Although both MOCVD and PEALD can produce high-quality films, reactive magnetron sputtering proves more advantageous due to its superior scalability and cost-effectiveness. Wang et al. [18] investigated the effects of substrate temperature and bias voltage on the properties of AlN thin films deposited on glass substrates by DC magnetron sputtering. Their study demonstrated that precise control of temperature and bias voltage parameters can effectively optimize both the crystallization process and deposition rate of aluminum nitride (AlN) films on glass substrates. Most existing methods employ high sputtering power, which intensifies ion bombardment energy during deposition. This enhanced energy increases the surface mobility of deposited atoms/molecules, potentially leading to irregular particle agglomeration, surface protrusions, and ultimately elevated surface roughness. This study investigates whether applying enhanced bias voltage under low-power reactive magnetron sputtering conditions can facilitate the preparation of AlN thin films with both low surface roughness and the preferred (002) orientation.

This research employs reactive magnetron sputtering to deposit AlN thin films on sapphire substrates, with particular emphasis on elucidating the influence of bias voltage on the structural characteristics of (002)-oriented AlN films. The findings aim to accumulate fundamental data for optimizing high-quality AlN thin film fabrication.

## 2. Materials and Methods

### 2.1. Sample Preparation

Sapphire 2-inch with C-axis orientation was selected as the substrate (Surface Roughness < 0.3 nm). Before magnetron sputtering, the substrate should be processed. First, the sapphire sheet should be immersed in acetone for ultrasonic cleaning for 15 min, then placed into anhydrous ethanol for ultrasonic cleaning for 10 min, and then placed into deionized water for ultrasonic cleaning for 5 min to ensure that the surface of the sapphire sheet is cleaned, before it is dried with high-purity nitrogen for use. In this paper, the equipment used for preparing AlN thin films is the TRP450 high-vacuum magnetron sputtering deposition system developed and produced by Shenyang Scientific Instruments Co., Ltd., Chinese Academy of Sciences, Shenyang, China. Its internal structure is shown in Figure 2, with a heating base on the top and one RF target and two DC targets installed on the bottom. In this experiment, A 2-inch diameter planar high-purity aluminum (99.99%) target was used as the RF magnetron sputtering source, the sputtering gas was high-purity argon (99.99%), and the reaction gas was high-purity nitrogen (99.99%).

First, the vacuum degree in the chamber was pumped below 1.0 × 10^−4^ Pa and the base temperature was raised to 780 °C. Then, N_2_ and Ar were injected; the chamber pressure was 1.9 Pa. The purpose of this was to remove the impurities on the surface of the target for 5 min. Finally, the shielding plate was opened to start the formal sputtering. Other parameters are shown in Table 1.

### 2.2. Sample Test

In this study, SmartLab series X-ray diffractometer (XRD) produced by Rigaku Corporation of Japan, Tokyo, Japan, was used to analyze the crystal condition and phase structure of AlN thin films. The surface roughness of AlN films was measured using an atomic force microscope (AFM) model FM-NanoviewOp-AFM, produced by Suzhou Flyingman Precision Instruments Co., Ltd. in Suzhou, China. The surface and cross-section morphology of the sample were observed by the scanning electron microscope (SEM) of German ZEISS Sigma 360, Baden-Württemberg, Germany.

## 3. Results and Discussion

All data are based on a single film growth attempt for each condition and should therefore be considered preliminary.

### 3.1. Topography Analysis

Figure 3a–f demonstrate the surface morphologies of aluminum nitride (AlN) thin films under different bias voltage conditions. The experimental results reveal that when the applied bias voltage exceeds 0 V, the AlN surface particles exhibit a dense and uniform configuration. This phenomenon can be attributed to the following mechanism: under zero bias conditions, the sputtered Al atoms possess insufficient kinetic energy, resulting in limited surface diffusion capability of the generated AlN particles. This energy deficiency prevents effective particle rearrangement and densification during film growth. However, under positive bias conditions (V > 0), the enhanced ion bombardment effect provides additional energy to the adatoms, facilitating their surface migration and subsequent formation of compact microstructures. Notably, variations in bias magnitude within the positive voltage regime do not induce significant modifications to the surface topography, suggesting a saturation effect in energy transfer beyond a critical bias threshold.

Figure 4a–f present cross-sectional morphologies of six sets of aluminum nitride (AlN) thin films. The images reveal that the AlN films exhibit a columnar crystal structure under the experimental conditions of this study. The film growth rates, calculated by averaging multiple thickness measurements of cross-sectional profiles, are displayed in Figure 5. The corresponding growth rates are 2.61 nm/min, 2.69 nm/min, 2.75 nm/min, 2.48 nm/min, 2.42 nm/min, and 2.33 nm/min, respectively. Notably, the growth rate demonstrates an initial increase followed by a decrease with ascending bias voltage. This trend aligns with the observations in Ref. [18].

This dual behavior can be rationalized through plasma–surface interaction dynamics. In the 0–40 V range, enhanced particle energy from intensified bias strengthens the sputtering yield, thereby increasing the flux of Al atoms/molecules ejected from the target per unit time. This elevated particle flux directly contributes to the observed growth-rate enhancement. However, within the 40–100 V regime, excessive ion bombardment energy induces two counteracting effects—while initial deposition continues, high-energy incident particles transfer sufficient momentum to surface-adhered species, triggering the re-sputtering of previously deposited material [19]. This competitive process between deposition and re-sputtering results in net growth rate attenuation at elevated bias voltages.

Figure 6a–f present atomic force microscopy (AFM) images of AlN thin films deposited under varying bias voltages. Figure 7 illustrates the relationship between the root mean square (RMS) roughness of AlN films and applied bias voltage, with measured values of 282 pm, 225 pm, 204 pm, 194 pm, 196 pm, and 167 pm, respectively. The minimum RMS roughness (167 pm) was achieved at 100 V bias voltage. A general downward trend in RMS roughness was observed with increasing bias voltage, which is consistent with previous study [2]. This phenomenon is attributed to the insufficient surface migration energy of atomic clusters at lower bias voltages, which promotes cluster-like growth morphology and consequently elevates surface roughness. As the bias voltage increases, enhanced energy transfer to adatoms facilitates improved surface diffusion and atomic rearrangement, leading to denser film morphology and reduced surface roughness [20]. 

### 3.2. Structural Analysis

Figure 8 displays the X-ray diffraction (XRD) patterns of AlN thin films under different bias voltage conditions. The crystallite sizes were calculated using the Scherrer formula:$$D=\frac{K\cdot \lambda }{\beta \cdot cos\;\theta }$$
where D represents the crystallite size, K is the Scherrer constant (0.89), is the X-ray wavelength (0.154 nm), β denotes the full width at half maximum (FWHM) in radians, and θ is the Bragg diffraction angle. For the AlN(002) plane (θ = 36.1°), the calculated crystallite sizes at bias voltages of 0 V, 20 V, 40 V, 60 V, 80 V, and 100 V are 103 nm, 110 nm, 111 nm, 112 nm, 113 nm, and 116 nm, respectively, as illustrated in Figure 9. This finding differs from the phenomenon observed in Reference [2], where grain size remained essentially stable with increasing bias voltage. Our study reveals a clear trend of increasing AlN grain size with higher bias voltages. This discrepancy suggests that, unlike deposition on conventional glass substrates, the application of elevated bias voltages during AlN film growth on sapphire substrates can effectively enhance grain growth.

The observed trend may be attributed to the energy-dependent formation mechanism of B2 covalent bonds along the c-axis orientation in AlN(002) crystals [21]. Elevated bias voltages enhance the kinetic energy and chemical reactivity of sputtered Al atoms, promoting more efficient bonding with nitrogen ($N_{2}$) species during reactive sputtering. This energy-driven optimization of atomic incorporation facilitates grain-coarsening. Furthermore, the diffraction peak intensity exhibits a non-monotonic behavior—initial the increase, subsequent decrease, and final recovery—which correlates with the previously reported growth rate variations. The reduced peak intensity at 60 V likely stems from diminished film thickness caused by growth-rate suppression under intermediate bias conditions, consistent with the deposition kinetics discussed earlier. 

## 4. Conclusions

(1) The growth rate of the AlN thin film exhibits a non-monotonic dependence on the applied bias voltage, showing an initial increase followed by a subsequent decrease. Specifically, the growth rate rises from 2.61 nm/min at 0 V to a maximum of 2.75 nm/min at 40 V. However, as the bias voltage is further increased to 100 V, the growth rate gradually declines to 2.33 nm/min. 

(2) The enhancement of the applied bias voltage led to an overall reduction in the RMS surface roughness of the films, decreasing from 282 pm at 0 V to 167 pm at 100 V, thereby effectively improving film quality through surface planarization.

(3) X-ray diffraction analysis reveals that as the bias voltage increases from 0 V to 100 V, the grain size grows from 103 nm to 116 nm. The film deposited at 100 V bias demonstrates the highest diffraction peak intensity and largest grain size, indicating optimal crystalline quality at this condition.

(4) While 100 V bias yielded optimal film quality in this study, further increases in bias voltage may introduce excessive ion bombardment, potentially leading to defects or amorphization.

## Figures and Tables

**Figure 1 micromachines-16-01027-f001:**
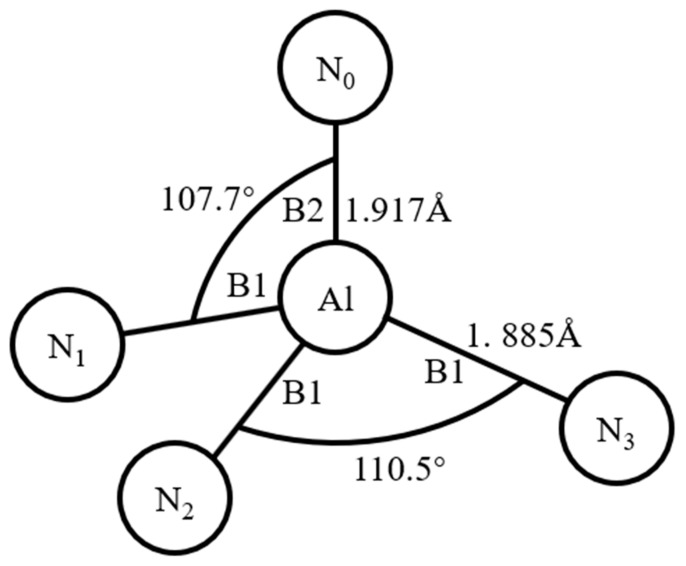
Bond angle schematic of AlN.

**Figure 2 micromachines-16-01027-f002:**
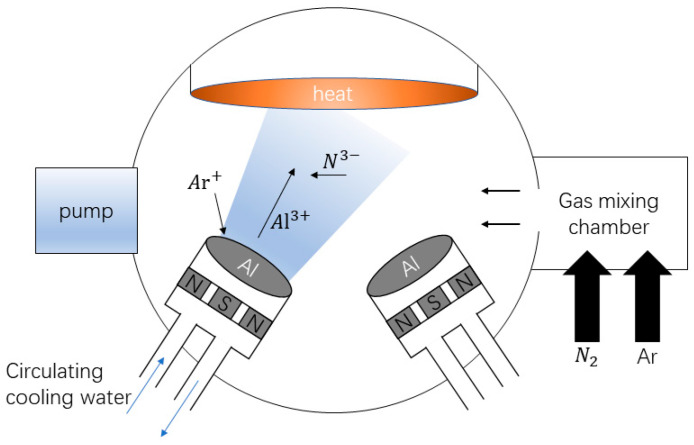
Schematic diagram of the equipment.

**Figure 3 micromachines-16-01027-f003:**
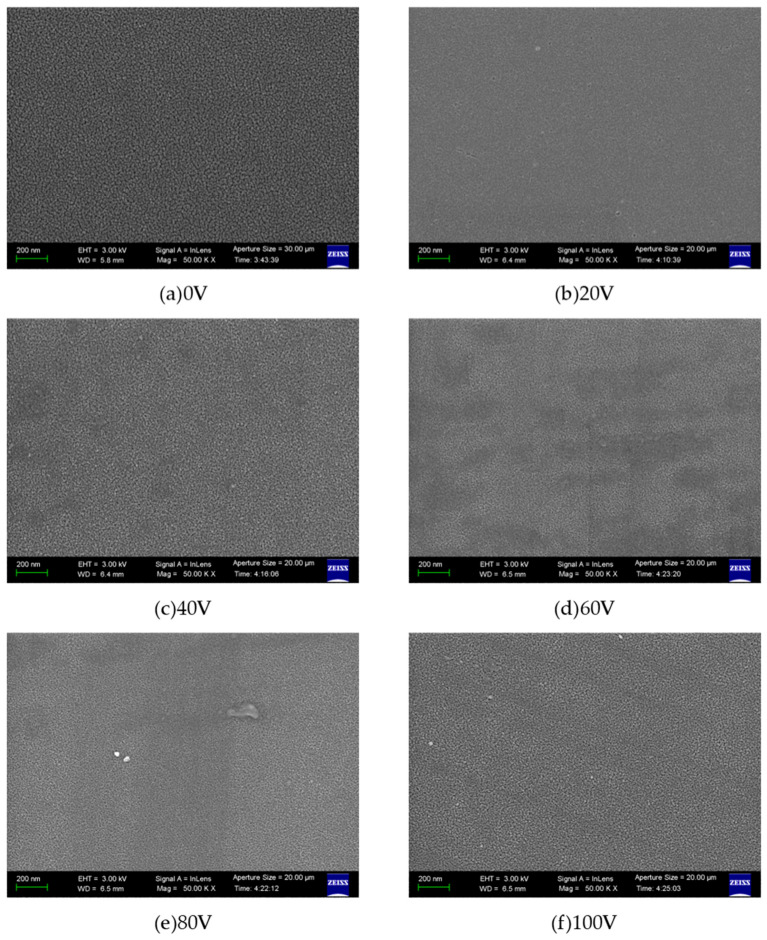
AlN surface topography images under different bias pressures.

**Figure 4 micromachines-16-01027-f004:**
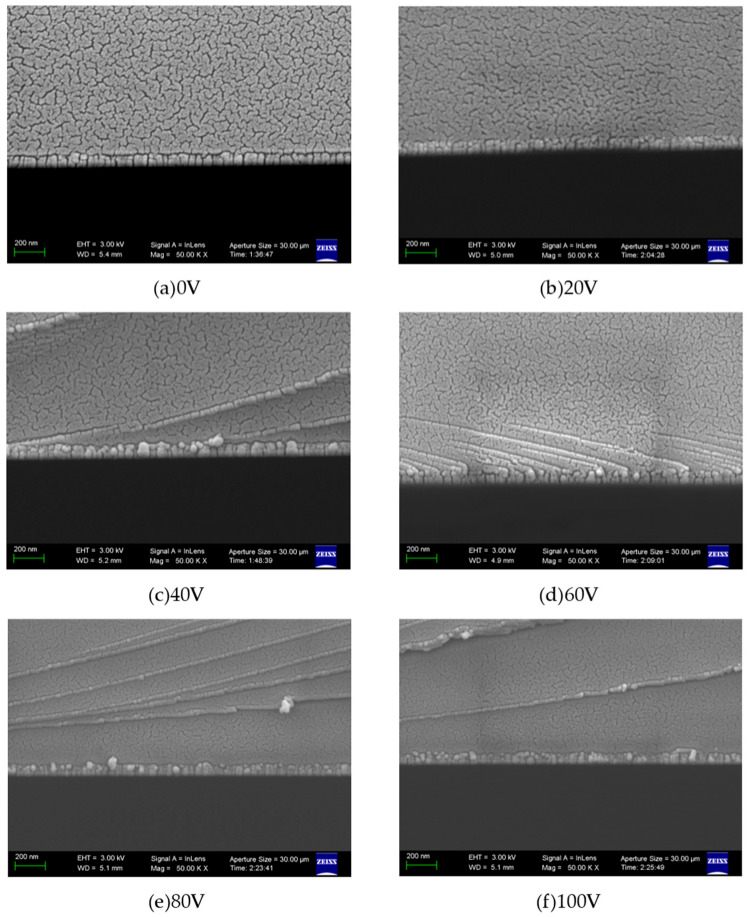
SEM images of AlN cross-section under different bias pressures.

**Figure 5 micromachines-16-01027-f005:**
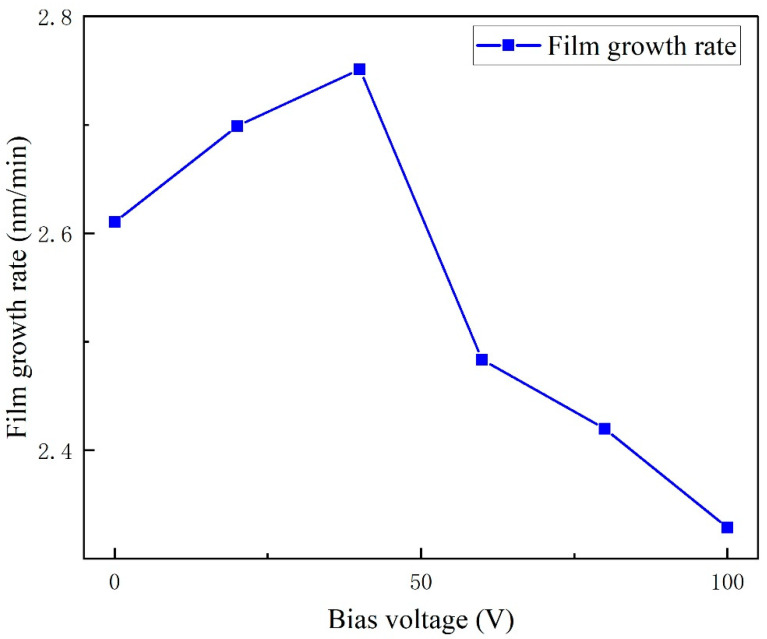
Relationship between AlN film growth rate and bias pressure.

**Figure 6 micromachines-16-01027-f006:**
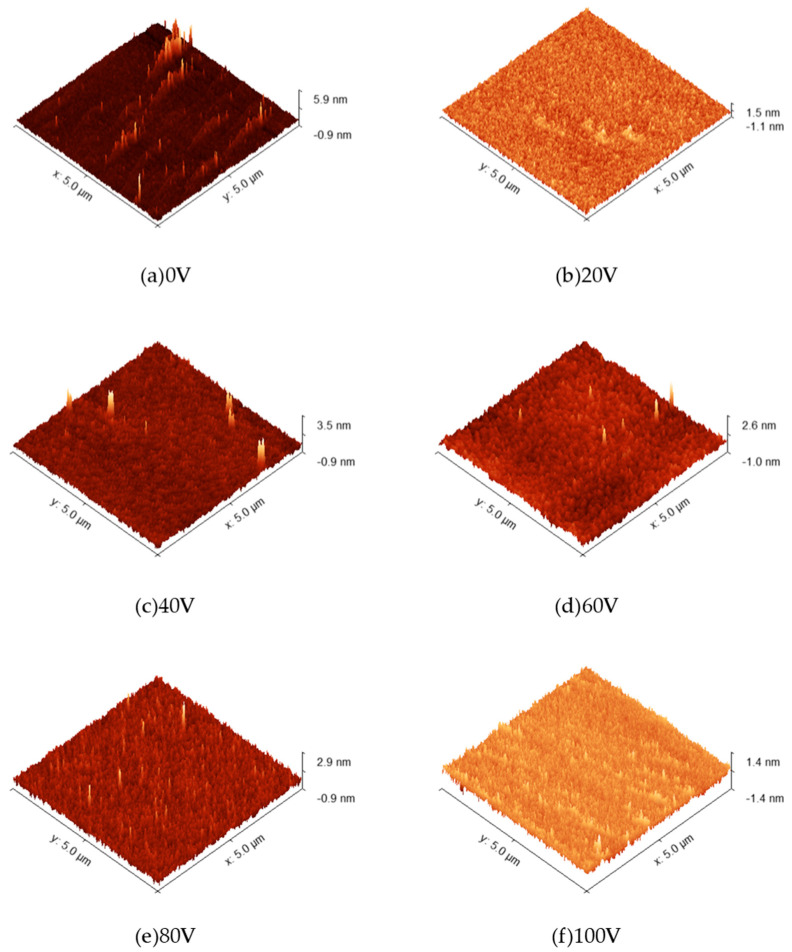
AFM test images of AlN thin films under different bias conditions.

**Figure 7 micromachines-16-01027-f007:**
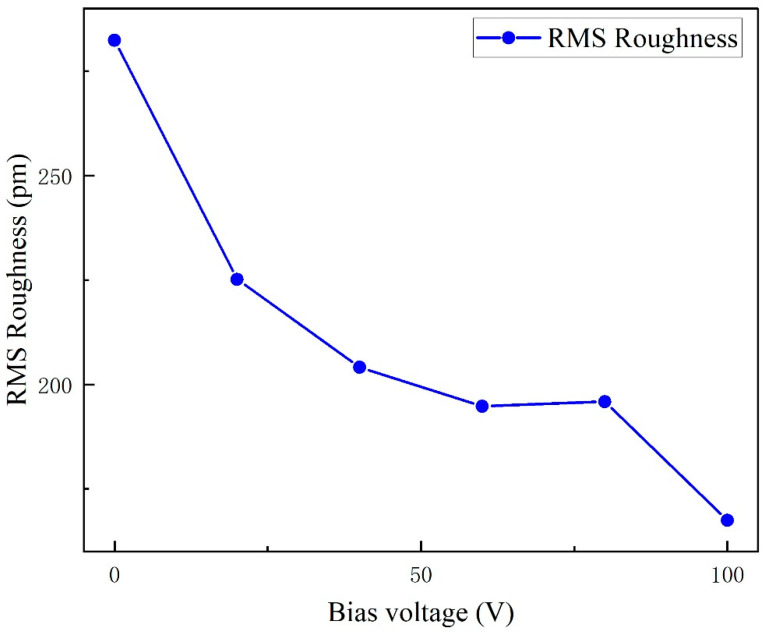
AlN film root–mean–square roughness and bias.

**Figure 8 micromachines-16-01027-f008:**
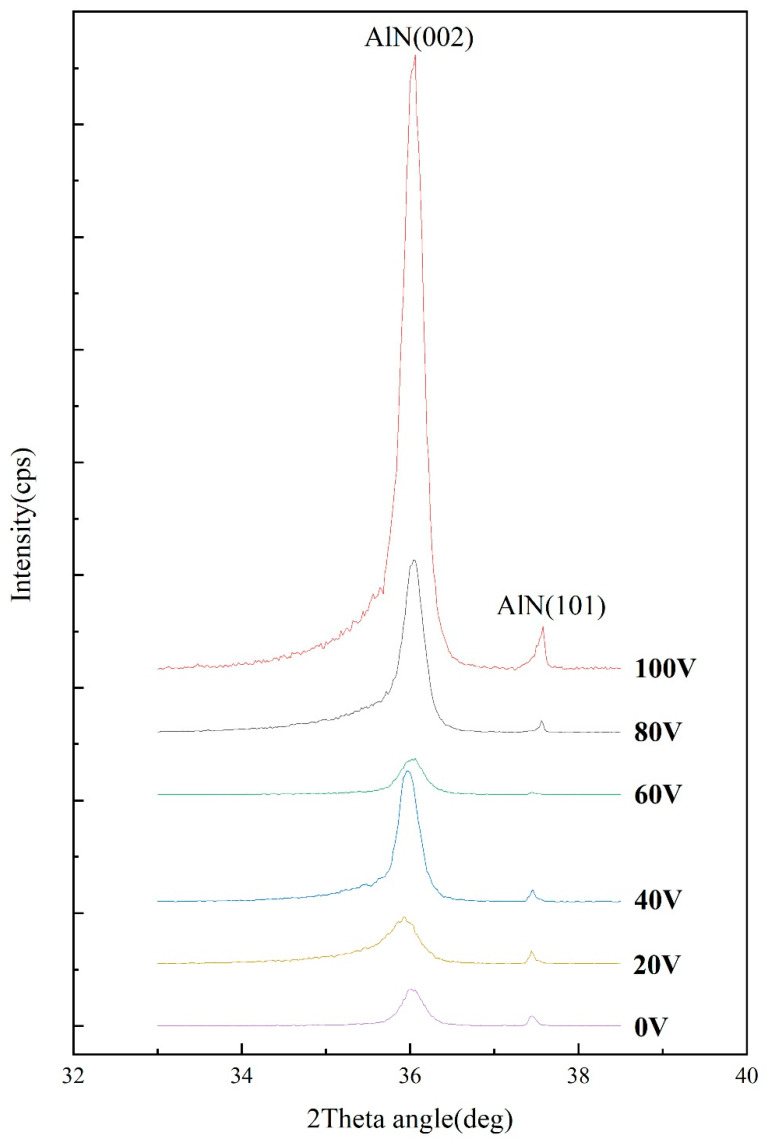
XRD pattern of AlN thin films at different bias pressures.

**Figure 9 micromachines-16-01027-f009:**
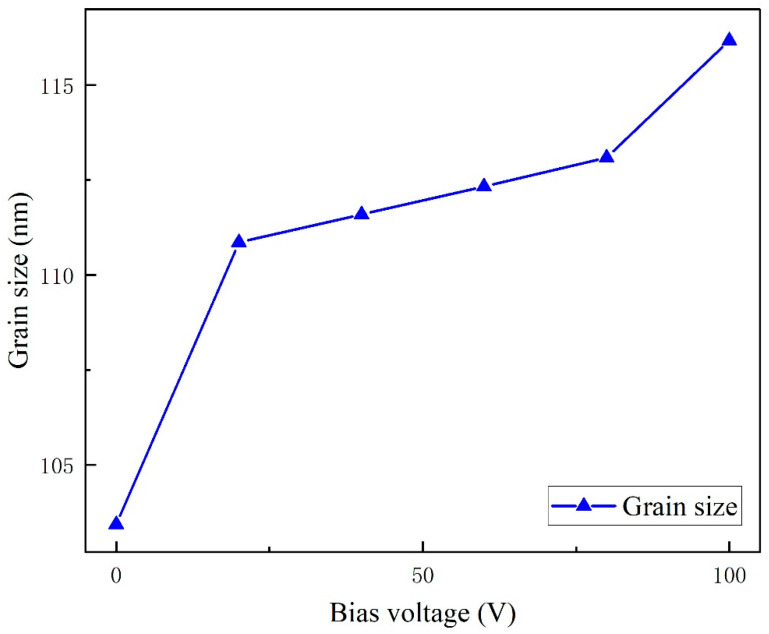
Grain size of AlN films at different bias pressures.

**Table 1 micromachines-16-01027-t001:** Sputtering parameters of each AlN sample.

Number	RF Power (W)	Bias Voltage (V)	$N_{2}$ Flow (sccm)	Ar Flow (sccm)	Sputtering Time (min)
1	200	0	10	10	30
2	200	20	10	10	30
3	200	40	10	10	30
4	200	60	10	10	30
5	200	80	10	10	30
6	200	100	10	10	30

## Data Availability

The original contributions presented in this study are included in the article.

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
