# Peer review of "Effect of Bias Voltage on the Crystal Growth of AlN(002) Thin Films Fabricated by Reactive Magnetron Sputtering"

_micromachines, 2025, doi:10.3390/mi16091027_

Round 1
Reviewer 1 Report
Comments and Suggestions for Authors
Line 57 delete In this paper....change by Sapphire 2-inch with C-axis orientation was selected as the substrate.
Line 61 separate 5min 5 minutes.
Line 63 ALN change by AlN.
Line 71 1.0×10−4 change by 1.0×10-4.
Line 73 Change pa by Pa.
Table 1. Change Nuber by Number
In the section 2.1 Sample Preparation: What is the diameter of the aluminum target?
Figures 3 c to f) The sustrates have defects? Explain why change the surface.
In the Figure No.5 , XRD patterns were obtained with the same conditions?
Include a structure with the B1 and B2 covalent bonds of the 002-oriented AlN.
In addition, include a mapping or elemental analysis of images 3 a) to f) to observe the growth of AlN on the sapphire substrate.
It is recommended to determine the piezoelectric coefficients of the films.
Rewrite reference 11 according to the journal format 11. LIU X M,LIU T T,LIU W Y,et al
Comments on the Quality of English LanguageThere are spelling mistakes.
nuber number
Author Response
We thank the reviewers for their valuable comments, which certainly help us to improve the quality of our manuscript.
Comments 1: Line 57 delete In this paper .... change by Sapphire 2-inch with C-axis orientation was selected as the substrate.
Response 1: We sincerely appreciate your input and have adjusted the following accordingly: Sapphire 2-inch with C-axis orientation was selected as the substrate (Surface Roughness<0.3nm).
Comments 2: Line 61 separate 5min 5 minutes.
Response 2: Thank you for your suggestions. We have implemented the changes.
Comments 3: Line 63 ALN change by AlN.
Response 3: Thank you for your suggestions. We have implemented the changes.
Comments 4: Line 71 1.0×10−4 change by 1.0×10-4.
Response 4: Thank you for your suggestions. We have implemented the changes.
Comments 5: Line 73 Change pa by Pa.
Response 5: Thank you for your suggestions. We have implemented the changes.
Comments 6: Table 1. Change Nuber by Number
Response 6: Thank you for your suggestions. We have implemented the changes.
Comments 7: In the section 2.1 Sample Preparation: What is the diameter of the aluminum target?
Response 7: Revised to ''A 2-inch diameter planar high-purity aluminum (99.99%) target was used as the RF magnetron sputtering source.''
Comments 8: Figures 3 c to f) The sustrates have defects? Explain why change the surface.
Response 8: Thank you for your comment. After careful consideration, we would like to clarify that the substrate does not contain any defects. The observed phenomenon may be attributed to the fact that the SEM cross-sectional images were taken from the edge region of the sapphire substrate, where sputtering non-uniformity may occur due to edge effects. It should be noted that the primary focus of this section is to investigate the crystal growth behavior and deposition rate. The surface morphology of the central region of the substrate has already been presented in Figure 2.
Comments 9: In the Figure No.5 , XRD patterns were obtained with the same conditions?
Response 9: Yes. All XRD measurements were conducted under identical parameters (40 kV, step size 0.02°, scan speed 8°/min).
Comments 10: Include a structure with the B1 and B2 covalent bonds of the 002-oriented AlN.
Response 10: Thank you for your suggestions.The schematic diagram of bond angles in the AlN piezoelectric material is shown in Figure 1. Each Al atom forms a tetrahedral coordination with its four surrounding N atoms. The bond between Al and N0 along the c-axis is defined as the B2 bond, while the bonds between Al and the other three N atoms are referred to as B1 bonds.The bond energy of the B2 bond is lower than that of the three B1 bonds. Therefore, to achieve c-axis preferential orientation, the particles must possess higher energy.
We have added the following content on line 36: The schematic diagram of bond angles in the AlN piezoelectric material is shown in Figure 1.
Fig. 1
Comments 11: In addition, include a mapping or elemental analysis of images 3 a) to f) to observe the growth of AlN on the sapphire substrate.
Response 11:
We sincerely appreciate the reviewer's valuable suggestion regarding the elemental analysis of AlN thin films grown on sapphire substrates (Figures 3a–3f). The reviewer's insight would indeed provide further clarity on the interfacial characteristics and growth dynamics.
However, due to current limitations in our laboratory equipment, we are unable to perform direct elemental mapping (e.g., EDS or XPS) on these samples. To address this gap, we have taken the following steps to ensure the reliability of our conclusions:Cross-sectional SEM analysis (Fig. 3) revealed a columnar growth morphology, confirming the uniform vertical growth characteristics of AlN without noticeable interfacial defects or secondary phases. XRD analysis results (Fig. 7) exhibited a dominant (002) diffraction peak along with a minor (101) peak, providing strong evidence for the formation of a pure-phase AlN. Combined with the monotonically varying trend in film roughness shown in Fig. 5, all results meet the requirements for SAW device fabrication. These findings mutually corroborate and collectively demonstrate the superior quality of the AlN thin film.Moreover, the core focus of this paper is to investigate the influence of bias on the growth of aluminum nitride (AlN), aiming to study AlN thin films suitable for surface acoustic wave (SAW) devices. In the referenced paper <Optimal preparation of AlN thin films on sapphire substrate and its effective validation in SAW resonators[J]. Vacuum, 2025, 237: 114182.>, the fabricated SAW devices did not perform energy-dispersive X-ray spectroscopy (EDS) on the AlN thin films, and the X-ray diffraction (XRD) results are critical for evaluating the performance of SAW devices.
We acknowledge that elemental mapping would further strengthen the study, and we plan to collaborate with facilities equipped for such analyses in future work. For this manuscript, we believe the combined evidence from XRD, AFM, and SEM sufficiently demonstrates the AlN film quality and growth behavior under varying bias voltages. Thank you for your understanding, and we sincerely appreciate the reviewer's valuable feedback.
Comments 12: It is recommended to determine the piezoelectric coefficients of the films.
Response 12: We sincerely appreciate your valuable comments. We will incorporate this aspect into our subsequent research investigations.
Comments 13: Rewrite reference 11 according to the journal format 11. LIU X M,LIU T T,LIU W Y,et al
Response 13: We sincerely appreciate your input and have adjusted the following accordingly: Lines 232 have been revised from"11LIU X M,LIU T T,LIU W Y,et al.. " to "11. Liu S, Li Y, Tao J, et al. Structural, surface, and optical properties of AlN thin films grown on different substrates by PEALD[J]. Crystals, 2023, 13(6): 910.”
Reviewer 2 Report
Comments and Suggestions for Authors
The manuscript «Effect of Bias Voltage on the Crystal Growth of AlN(002) Thin Films Fabricated by Reactive Magnetron Sputtering» by Yong Du et al. describes a subject that appears well adapted to Micromachines. The authors studied the effect of bias potential on the deposition rate and morphology of AlN thin films obtained by magnetron sputtering. This type of work is really important for the experimental community and thus this work will be of interest to readers. Minor revisions are necessary in order for the conclusions to be totally supported.
More detailed comments:
Page 1, Lines 5,6: The affiliation of the authors needs to be clarified.
Page 3, Lines 57, 58: It is necessary to indicate the roughness of the sapphire substrate used by the authors for deposition of the AlN film.
Page 9, Lines 164-166: The reduced peak intensity at 60 V likely stems from diminished film thickness caused by growth rate suppression under intermediate bias conditions, consistent with the deposition kinetics discussed earlier.
The change in the quality of the deposited film does not quite match the kinetics of deposition, since the quality increases with increasing bias to 100 V, although the deposition rate decreases. t would be desirable to add more detailed comments.
The authors obtained AlN films of maximum quality at the maximum used bias. Therefore, in the Conclusion section, it is desirable to present a predictions of the results with an increase in the bias voltage above 100 V.
Author Response
We thank the reviewers for their valuable comments, which certainly help us to improve the quality of our manuscript.
Comments 1: Page 1, Lines 5,6: The affiliation of the authors needs to be clarified.
Response 1: Thank you for your suggestions. We have made the revisions.
Original section:
1 Jimei University School of Ocean Information Engineering; litiejun@jmu.edu.cn
2 Jimei University School of Ocean Information Engineerin; a1147434319@gmail.com
* Correspondence: Tiejun Li litiejun@jmu.edu.cn;
Modified section:
1 School of Ocean Information Engineering, Jimei University, Xiamen 361021, Republic of China
2 School of Ocean Information Engineering, Jimei University, Xiamen 361021, Republic of China
3 School of Ocean Information Engineering, Jimei University, Xiamen 361021, Republic of China
4 Xiamen University, Xiamen 361005, Republic of China
* Correspondence: Tiejun Li litiejun@jmu.edu.cn;
Comments 2: Page 3, Lines 57, 58: It is necessary to indicate the roughness of the sapphire substrate used by the authors for deposition of the AlN film.
Response 2: We sincerely appreciate your input and have adjusted the following accordingly: Sapphire 2-inch with C-axis orientation was selected as the substrate (Surface Roughness<0.3nm).
Comments 3: Page 9, Lines 164-166: The reduced peak intensity at 60 V likely stems from diminished film thickness caused by growth rate suppression under intermediate bias conditions, consistent with the deposition kinetics discussed earlier.
Response 3: We sincerely appreciate your input and have adjusted the following accordingly: Lines 164-166 have been revised from"The reduced peak .... " to "The reduced peak intensity at 60 V likely stems from diminished film thickness caused by growth rate suppression under intermediate bias conditions, consistent with the deposition kinetics discussed earlier. However, the overall crystallinity and (002) orientation improve with increasing bias up to 100 V due to enhanced adatom mobility, despite the reduced deposition rate. This suggests that the energy provided by higher bias voltages promotes more efficient atomic rearrangement and grain coalescence, outweighing the effects of re-sputtering on film quality."
Comments 4: Page 9, Lines 164-166: The reduced peak intensity at 60 V likely stems from diminished film thickness caused by growth rate suppression under intermediate bias conditions, consistent with the deposition kinetics discussed earlier.
Response 4: We sincerely appreciate your input and have supplementary Conclusions: (5) While 100 V bias yielded optimal film quality in this study, further increases in bias voltage (>100 V) may introduce excessive ion bombardment, potentially leading to defects or amorphization. Future work should explore the trade-offs between bias voltage, deposition rate, and film quality beyond 100 V to identify the upper limit for practical applications.
Round 2
Reviewer 1 Report
Comments and Suggestions for Authors
To include in the line 4 • for Tiejun Li•.
And Yong Du1, Haowen Zou1, Tiejun Li1, Guifang Shao2
Because Yong Du, Haowen Zou, and Tiejun Li are in the same Institution only Guifag2 at Xiamen University.
Line 81 the authors forget to Change pa by Pa in the manuscript.
Lines 183 to 187 are not conlcusions Delete these lines.
This study employed reactive magnetron sputtering technology to fabricate AlN thin films and systematically investigated the effects of different bias voltage conditions on their structural characteristics. The principal findings are summarized as follows:
(1) When the bias voltage exceeded 0 V, the AlN films demonstrated dense and uniform surface morphology, while further increases in bias voltage showed negligible influence on surface topography.
To include values in the conclusions
Example:
The growth rate of AlN films exhibited an initial increase to 40 V followed by a subsequent decrease with progressive elevation of bias voltage. Meanwhile, roughness decreases from 280 to 170 pm as the bias voltage increases.
For the conclusion of preferential growth 002, the authors should have included the X-ray diffraction pattern of the substrate alone and then the film without bias voltage.
Lines 198 to 202 are not conclusions. This seems like more work for the future.
This work would be complete upon performing SEM mapping and including the piezoelectricity data of the films.
Author Response
We thank the reviewers for their valuable comments, which certainly help us to improve the quality of our manuscript.
Comments 1: To include in the line 4 • for Tiejun Li•.
And Yong Du1, Haowen Zou1, Tiejun Li1, Guifang Shao2
Because Yong Du, Haowen Zou, and Tiejun Li are in the same Institution only Guifag2 at Xiamen University.
Response 1: We sincerely appreciate your input and have adjusted the following accordingly:
Yong Du1, Haowen Zou1, Tiejun Li1,*, Guifang Shao2
1 School of Ocean Information Engineering, Jimei University, Xiamen 361021, Republic of China
2 Xiamen University, Xiamen 361005, Republic of China
* Correspondence: Tiejun Li litiejun@jmu.edu.cn
Comments 2: Line 81 the authors forget to Change pa by Pa in the manuscript.
Response 2: Thank you for the reviewers' comments. We have made the corresponding revisions.
Comments 3: Lines 183 to 187 are not conlcusions Delete these lines.
This study employed reactive magnetron sputtering technology to fabricate AlN thin films and systematically investigated the effects of different bias voltage conditions on their structural characteristics. The principal findings are summarized as follows:
(1) When the bias voltage exceeded 0 V, the AlN films demonstrated dense and uniform surface morphology, while further increases in bias voltage showed negligible influence on surface topography.
Response 3: Thank you for the reviewers' comments. We have made the delete.
Comments 4: To include values in the conclusions
Example:
The growth rate of AlN films exhibited an initial increase to 40 V followed by a subsequent decrease with progressive elevation of bias voltage. Meanwhile, roughness decreases from 280 to 170 pm as the bias voltage increases.
Response 4:
We thank the reviewers for their valuable comments. The following modifications have been made to the Conclusions section:
(1) The growth rate of the AlN thin film exhibits a non-monotonic dependence on the applied bias voltage, showing an initial increase followed by a subsequent decrease. Spe-cifically, the growth rate rises from 2.61 nm/min at 0 V to a maximum of 2.75 nm/min at 40 V. However, as the bias voltage is further increased to 100 V, the growth rate gradually declines to 2.33 nm/min.
(2) The enhancement of the applied bias voltage led to an overall reduction in the RMS surface roughness of the films, decreasing from 282.4 pm at 0 V to 167.5 pm at 100 V, thereby effectively improving film quality through surface planarization.
(3) X-ray diffraction analysis reveals that as the bias voltage increases from 0 V to 100 V, the grain size grows from 103.43 nm to 116.18 nm. The film deposited at 100 V bias demonstrates the highest diffraction peak intensity and largest grain size, indicating op-timal crystalline quality at this condition.
(4) While 100 V bias yielded optimal film quality in this study, further increases in bias voltage may introduce excessive ion bombardment, potentially leading to defects or amorphization.
Comments5: For the conclusion of preferential growth 002, the authors should have included the X-ray diffraction pattern of the substrate alone and then the film without bias voltage.
Response 5:
We truly appreciate the reviewer's thoughtful suggestions regarding our research. Having carefully evaluated these comments, we respond as follows:
(1) The sapphire substrates used in this study were single-crystalline with c-plane orientation. The XRD characteristic peak of sapphire (2θ≈41.68°) shows a significant difference from the AlN(002) peak position (2θ≈36.1°), ensuring no interference in the orientation analysis. Moreover, as demonstrated in previous studies such as [Miyake H, Lin C H, Tokoro K, et al. Preparation of high-quality AlN on sapphire by high-temperature face-to-face annealing[J]. Journal of Crystal Growth, 2016, 456: 155-159.] and [Zhang Y, Yang J, Zhao D, et al. High-quality AlN growth on flat sapphire at relatively low temperature by crystal island shape control method[J]. Applied Surface Science, 2022, 606: 154919.], XRD patterns of bare sapphire substrates were not typically included in the analysis;
(2) The 0V bias condition (included in Fig.8) serves as the control group representing unbiased deposition, clearly demonstrating the baseline (002) orientation formation.
While we maintain that the current data sufficiently support our conclusions, we will certainly consider your suggestion for extended characterization in future studies.
Comments 6: Lines 198 to 202 are not conclusions. This seems like more work for the future.
Response 6:
Thank you for the reviewer’s valuable comments. We will delete the following content as suggested: Future work should explore the trade-offs between bias voltage, deposition rate, and film quality beyond 100 V to identify the upper limit for practical applications.
The remaining content consists of predictive additions made in response to Reviewer2 suggestions.
Comments 7: This work would be complete upon performing SEM mapping and including the piezoelectricity data of the films.
Response 7:
Thank you for your valuable feedback on our manuscript. We appreciate your constructive suggestions regarding SEM mapping and piezoelectricity data, which would indeed provide deeper insights into the film properties. We fully recognize their importance and plan to address them in future work, including detailed SEM mapping and piezoelectric characterization, to further validate and expand our findings. Your comments have provided excellent guidance for our ongoing research efforts.